# *Bulinus senegalensis* and *Bulinus umbilicatus* Snail Infestations by the *Schistosoma haematobium* Group in Niakhar, Senegal

**DOI:** 10.3390/pathogens10070860

**Published:** 2021-07-08

**Authors:** Papa Mouhamadou Gaye, Souleymane Doucoure, Bruno Senghor, Babacar Faye, Ndiaw Goumballa, Mbacké Sembène, Coralie L’Ollivier, Philippe Parola, Stéphane Ranque, Doudou Sow, Cheikh Sokhna

**Affiliations:** 1Aix-Marseille Université, IRD, AP-HM, SSA, VITROME, 13005 Marseille, France; papamg91@hotmail.com (P.M.G.); goumballa2011@hotmail.fr (N.G.); Coralie.lollivier@ap-hm.fr (C.L.); philippe.parola@univ-amu.fr (P.P.); Stephane.ranque@ap-hm.fr (S.R.); cheikh.sokhna@ird.fr (C.S.); 2VITROME, Campus International IRD-UCAD de l’IRD, 1386 Dakar, Senegal; souleymane.doucoure@ird.fr (S.D.); bruno.senghor@yahoo.fr (B.S.); 3Institut Hospitalo-Universitaire (IHU)-Méditerranée Infection de Marseille, 13005 Marseille, France; 4Département Biologie Animale, Faculté des Sciences et Technique, UCAD de Dakar, 5005 Dakar, Senegal; mbacke.sembene@ucad.edu.sn; 5Department of Parasitology-Mycology, Faculty of Medicine, Pharmacy and Odontology, University Cheikh Anta Diop of Dakar, 5005 Dakar, Senegal; bfaye67@yahoo.fr; 6Department of Parasitology-Mycology, UFR Sciences de la Santé, Université Gaston Berger, 234 Saint-Louis, Senegal

**Keywords:** schistosomiasis, *Schistosoma haematobium*-group, *S. bovis*, *Bulinus senegalensis*, *Bulinus umbilicatus*, Dra1, RD-PCR, Niakhar

## Abstract

Thorough knowledge of the dynamics of *Bulinus* spp. infestation could help to control the spread of schistosomiasis. This study describes the spatio-temporal dynamics of *B. senegalensis* and *B. umbilicatus* infestation by the *Schistosoma haematobium* group of blood flukes in Niakhar, Senegal. Molecular identification of the *S. haematobium* group was performed by real-time PCR, targeting the Dra 1 gene in 810 samples of *Bulinus* spp. collected during the schistosomiasis transmission season in 2013. In addition to Dra 1 PCR, a rapid diagnostic-PCR was performed on a sub-group of 43 snails to discriminate *S. haematobium*, *S. bovis*, and *S. mattheei.* Out of 810 snails, 236 (29.1%) were positive for Dra 1 based on the PCR, including 96.2% and 3.8% of *B. senegalensis* and *B. umbilicatus,* respectively. Among the sub-group, 16 samples were confirmed to be *S. haematobium* while one was identified as a mixture of *S. haematobium* and *S. bovis*. Snails infestations were detected in all villages sampled and infestation rates ranged from 15.38% to 42.11%. The prevalence of infestation was higher in the north (33.47%) compared to the south (25.74%). Snail populations infestations appear early in the rainy season, with a peak in the middle of the season, and then a decline towards the end of the rainy season. Molecular techniques showed, for the first time, the presence of *S. bovis* in the *Bulinus* spp. population of Niakhar. The heterogeneity of snail infestations at the village level must be taken into account in mass treatment strategies. Further studies should help to improve the characterizations of the schistosome population.

## 1. Introduction

Schistosomiasis is a debilitating chronic disease affecting both humans and livestock [1]. It is the most important neglected tropical disease (NTDs), second only to malaria in terms of the public health impact [2,3]. Schistosomiasis affects over 240 million people in low and middle-income countries in tropical regions with more than 200,000 deaths a year worldwide [4]. Up to 90% of the disease burden rests on countries in Saharan Africa, with school-age children and young adults representing the population most at risk.

The urogenital form of the disease caused by *Schistosoma haematobium* is the most common schistosomiasis condition in sub-Saharan Africa. Around 600 million people are at risk of infection with 110 million cases [5] and 150,000 deaths per year [6]. The spread and level of disease transmission are closely linked to hydro-agricultural developments which create favorable ecological conditions in which snails, intermediate hosts of *Schistosoma* parasites, can thrive [7,8]. This situation is exacerbated by poor socio-environmental conditions such as the lack of drinking water, unimproved sanitation, and poor hygiene engendering human contact with snails, thus maintaining the disease transmission cycle. 

In Senegal, *S. haematobium* is endemic in some rural areas, particularly in the center of the country with a prevalence of 11% in Niakhar [9] and in the northern region where the prevalence can reach 95% in some villages [10]. The freshwater gastropod snails of the genus *Bulinus* act as intermediate hosts of parasites of the *haematobium*-group [11]. *Bulinus senegalensis* and *Bulinus umbilicatus* are the main species involved in the seasonal transmission of *Schistosoma haematobium* in the central region of Senegal, particularly in the Niakhar area [12,13]. In Niakhar, despite various mass treatment campaigns using praziquantel (PZQ) [13], schistosomiasis is still endemic with a heterogeneous level of transmission between different villages. This suggests the need to add supplementary control tools to the usual chemotherapy to effectively break durably the cycle of transmission. Studying the distribution, the abundance of snails and their infestation by different schistosome species could help to improve our understanding of the dynamics of schistosomiasis transmission [14]. In addition, a precise study of the species of parasites co-infesting *Bulinus* could also enable a better understanding of the evolutionary dynamics of the parasites, especially the emergence of hybrid strains [15,16,17,18].

Indeed, different species of the *Schistosoma haematobium* group can interact with the genus *Bulinus*, meaning that there is a possibility of hybridization between human and livestock schistosomes [1]. These aspects are particularly important to consider when developing adequate malacological control strategies and gaining a further understanding of the disease epidemiology. 

To date, despite the presence of large livestock populations, no hybrid parasites have been documented in Niakhar located in central Senegal contrast to the high prevalence of such strains described in the northern part of the country. In addition, there is little information on the circulation of livestock parasites such as *S. bovis* and *S. curassoni* infesting *Bulinus* snails in the center of Senegal. This situation could be explained by the lack of molecular diagnostic tools used in previous investigations in Niakhar [12]. Therefore, this study was undertaken to describe the spatial and temporal distribution of snails infested by various species of *haematobium*-group. 

## 2. Materials and Methods

### 2.1. Study Area

The study area of the Niakhar demographic survey site (DSS) is located in the region of Fatick (14°30 N to 16°30 W), 135 km east of Dakar [19]. The area includes 30 villages which were home to approximatively 43,000 inhabitants in 2013 [20] covering an area of 230 km^2^ with a population density of 214 inhabitants per km^2^ [21]. The climate is Sahelian-Sudanese [20] with minimums temperatures ranging from 20.6 °C to 21.1 °C while the maximum temperatures range between 35.8 °C and 36.8 °C [22]. The Niakhar area is characterized by a dry season lasting for 7 to 8 months, and a rainy season of 4 to 5 months (from June–July to October) [23]. From 2011 to 2013, the average annual rainfall was 611.4 mm. 

The snails examined during the present study were from a previous collection made during a large survey in 2013 that took place in ten villages in Niakhar: the villages of Sob, Sass njafaj, Ngalagne kop, Tukar, Puday, Logdir, Diohin, Godel, Gadiak and Ngangarlam (Figure 1). In these villages, swimming, water collection, laundry, washing of domestic animals, and fishing are the main activities where people come into contact with water. In the villages of Logdir, Ngalagne kop, Ngangarlam, Puday, Sass njafaj, and Sob, all water-related activities take place in ponds. Of these six villages, only Ngangarlam did not have access to drinking water at the time of the study. In the villages of Sass njafaj and Sob, ponds are rare and far from households [12]. The villages of Ngalagne kop and Gadiak each had four ponds while Tukar, Puday, and Logdir each had three temporary ponds. The Sob, Sass njafaj, Ngangarlam, Diohin and Godel sites had only one pond per village. The villages of Gadiak, Ngalagne Kop, Tukar, Ngangarlam, and Puday are located in the north of Niakhar while Godel, Logdir, Sass njafaj, Sob, and Diohin are situated in the south. *B. senegalensis* was distributed throughout the study area while *B. umbilicatus* is localized only in Ngangarlam village [12]. The snails were collected at the beginning (July), middle (August–September), and the end of the rainy season (October–November). All the snails collected during this survey in 2013 were stored for additional testing. However, due to the poor quality of the stored specimens, only a few samples were tested [12]. 

### 2.2. Molecular Analysis

#### 2.2.1. DNA Extraction 

Deoxyribonucleic acid (DNA was extracted from *B. senegalensis* and *B. umbilicatus* snails collected in the survey carried out in 2013 [12]. Cetyltrimethylammonium bromide (CTAB) method was used to extract genomic DNA from whole snail feet obtained after their dissection. The foot is more readily accessible and has primary sporocysts of schistosomes [24].

The foot of each individual snail was placed in a 1.5 mL Eppendorf tube and crushed in 200 μL of 2% CTAB to digest the proteins before the mixture was incubated at 65 °C for 1 h. After incubation, 200 μL of chloroform was added to separate nucleic acids from other undesirable constituents. The mixture was then centrifuged at 12,000 rpm for 5 min and the supernatant was recovered in another tube containing 200 µL of isopropanol to precipitate the DNA. After centrifugation for 15 min at 12,000 rpm, the isopropanol was emptied, and the tube was drained in absorbent paper. The same procedure was carried out with 200 μL of pure ethanol. The purified DNA was dried using a Speed-Vac concentrator and then resuspend in 200 μL of pure water. The DNA extracts were stored at −20 °C until use.

#### 2.2.2. Real-Time PCR (RT-PCR) 

The real-time PCR technique was used to detect the presence of parasites of the *Schistosoma haematobium*-group species in DNA extracts from the snails. The highly repetitive Dra1 sequence was selected as a target, which enabled the detection of the *S. haematobium*-group. The primers used (forward: 5′-GATCTCACCTATCAGACGAAAC-3′; reverse: 5′-TCACAACGATACGACCAAC-3′) were identical to those initially described by Hamburger et al. [25] and Ibironke et al. [26] and the probe (5′-TGTTGGTGAAGTGCCT GTTTCGCAA-3′) utilized was described by Cnops et al. [27]. The real-time PCR targeting the Dra1 gene was performed with a 19.5 μL reaction mixture containing 5 μL of DNA, 3 μL of sterile distilled water, 0.5 μL of each primer, 0.5 μL of TaqMan^TM^ (Applied Biosystems, Foster City, CA, USA) probe, and 10 μLof Master Mix. The reaction was carried out in CFX96 thermal cycler (Bio-Rad, Marnes-la-Coquette, France), with the program consisting of an initial denaturation step of 2 min at 50 °C followed by denaturation for 3 min at 95 °C, then 40 cycles of 95 °C for 30 s and 60 °C for 1 min before maintaining the sample at 4 °C. The result was considered as positive if the cycle threshold (Ct) value was less than 35. In each run, positive and negative controls were launched along with the samples.

#### 2.2.3. Rapid Diagnostic Multiplex PCR (RD-PCR) Discriminating *S. haematobium, S. bovis,* and *S. mattheei*

Positive samples obtained from Dra 1 real-time PCR, were tested by a discriminative multiplex PCR targeting *S. haematobium*, *S. bovis*, and *S. mattheei*. This PCR technique employed markers of different lengths in mitochondrial sub-unit 1 of the cytochrome c oxidase 1 (COX1) gene [28]. Amplification was achieved in a reaction volume of 25 μl including 2 μL of extracted snails DNA and 23 μl of Master Mix AmpliTaq Gold^®^ 360 PCR (Applied Biosystems™, Waltham, MA., USA). The concentrations of RD-PCR primer in the reaction were 0.2 μM of direct universal primer Asmit1 (TTTTTTGGTCATCCTGAGGTGTAT) and 0.2 μM of each reverse primer (Sh. R:5′-TGATAATCAATGACCCTGCAATAA-3′, Sb.R: 5′-CACAGGATCAGACAAACGAGTACC-3′ and Smat.R: 5′-CACCAGTTACACCACCAACAGA-3′, respectively [18,28]. The amplification protocol was carried out with initial denaturation at 95 °C for 15 min, with 39 cycles at 95 °C for 30 s, at 58 °C for 1 min, 72 °C for 1min and a final stage at 72 °C for 7 min using a Thermal Cycler (Applied Biosystems, Foster City, CA, USA). The migration was carried out for 1 h and 15 min to 180V in a 1.5% agarose gel with an SYBR safe dye and the reading was using a Gel Doc System (Bio-Rad, Hercules, CA, USA) (Figure 2).

#### 2.2.4. Statistical Analysis 

The data from the experiments were entered into the Microsoft Excel 2010 software and analyzed using R 3.6.3 software [29]. Statistical analyses were performed using the results of the prevalence of *haematobium* infestations in snails. Statistical analyses comparing the quantitative variables were carried out using the Pearson χ^2^ test with the R software. For all the analyses, a threshold of 5% was chosen as statistically significant. 

## 3. Results

### 3.1. Geographical Distribution of Snails Collected

A total of 810 snails were collected during the study [12] and tested for the presence of the *Schistosoma* parasite. The collection times of the snails were as follows: 134 (16.5%) in July, 399 (49.25%) in August-September, and 277 (34.19%) in October–November. Among these snails, *Bulinus senegalensis* comprised the majority of the sample with 790 individuals (97.5%) while *B. umbilicatus* was marginal with only 20 specimens collected (2.5%). The 810 snails tested were distributed in the villages as follows: Sob (50), Sass njafaj (38), Ngalagne kop (156), Tukar (130), Puday (113), Logdir (78), Diohin (49), Godel (30), Gadiak (116) and Ngangarlam (50). *B. umbilicatus* was identified only in Ngangarlam as indicated in Table 1.

### 3.2. Molecular Detection of the Haematobium-Group in the Snails 

A total of 810 DNA extracts from individual snail were tested by real-time PCR to detect *Schistosoma haematobium* species. In this sample, 236 snails were positive for the *haematobium*-group, representing an infestation rate of 29.1% (236/810). Among these positive snail specimens, 227 (96.2%) and 9 (3.8%) belonged to *B. senegalensis* and *B. umbilicatus*, respectively. Thus, the total prevalence of infestation in the *Bulinus* population was 28.7% (227/790) in *B. senegalensis* and 45.0% (9/20) in *B. umbilicatus*.

To further develop our understanding discriminative PCR able to differentiate *S. haematobium* into *S. bovis* and *S. mattheei*, was carried out on 43 samples positives for Dra1 through real-time PCR. These samples came from the villages of Sob (3), Sass Ndiafadj (3), Ngalagne kop (6), Toucar (5), Pouday (5), Logdir (2), Diohine (3), Godel (5), Gadiak (6), and Ngangarlam (5). Of the 43 samples only 16 (37.2%) were positive including 15 *S. haematobium* (543 bp) and 1 sample showing a double profile of *S. haematobium*/*S. bovis* (543 bp and 306 bp) (Figure 2). The remaining 28 sample results (65%) were negative according to the RD-PCR (Figure 2).

### 3.3. Dynamic of Snail Infestation 

#### 3.3.1. Infestation Levels of the Villages

##### Infestation Level per Site

The proportions of snails infested in villages are presented in Table 2. The highest infestation proportions were seen in Sass njafaj (42.11%) and Ngangarlam (42%). In Ngangarlam village where both snail species were found, the level of infestation within each species were at 45.0% (9/20) and 40.0% (12/30) in *B. umbilicatus* and *Bulinus senegalensis,* respectively (χ^2^ = 0.003, df = 1, *p* = 0.95 > 0.05).

##### Infestation According to the Rainy Season Period

The snail infestation rates were 27.61%, 37.09%, and 18.41% at the beginning, middle, and end of the rainy season, respectively. There were a significant difference between the snail infestations seen at the beginning, the middle, and the end of the season (χ^2^= 27.816, df = 2, *p* = 9.118 × 10^−7^). The infestation levels in villages were different according to the period of the rainy season. In some villages including Ngalagne Kop, Puday, and Diohin, transmission occurred at beginning of the rainy, reached a peak in the middle, and declined by the end of the rainy season. In Sob and Gadiak villages, the peak was noticed in the middle of the rainy season, with no snail infested at the beginning of the rainfall in July. At the same time, the peaks of infestation were observed in Logdir and Ngangarlam. In Logdir, the level of snail infestation (83.3%) found in July was the highest observed in the study site but dropped considerably in the middle and the end of the rainy season. In Ngangarlam *B. umbilicatus* and *B. senegalensis* were present in snails at the beginning and the middle of the season but, only *B. senegalensis* was present at the end of the season. In Godel, a snail infestation was only observed at the beginning of the rainy season. In Sass njafaj, the snail infestations occurred in the middle and at the end of the rainy season (Table 2) (Figure 3). Overall, in the majority of villages, snail infestations were observed very early in July; however, the infestation peaks occurred in the middle of the rainy season (Figure 3). The infestation rate was higher (33.47%) in the northern villages (Gadiak, Ngalagne Kop, Tukar, and Puday) compared to the proportion of the 25.74% observed in the southern villages (Godel, Logdir, Sass njafaj, Sob, and Diohin).

## 4. Discussion

The molecular analysis revealed a high infestation prevalence by trematodes of the *S. haematobium*-group in Niakhar (29.1%). A previous study in the area which used the same sampling technique, revealed an infestation rate of 0.5% to 0.8% using the cercarial shedding method [13]. This level of infestation is much lower than the proportion observed in our study. Similar situations were observed in *Schistosoma* endemic foci when molecular techniques are used to screen the presence of parasites in intermediate snail hosts [30,31,32]. This result demonstrates the need for more powerful diagnostic tools such as molecular biology to better assess the levels of infestation. In addition, molecular techniques are particularly important to discriminate the presence of human, livestock, and hybrids species [17,18]. Despite a low sampling rate, our RD-PCR study has shown us that *Schistosoma haematobium* is the main species found in Niakhar. This is in accordance with the epidemiological data in Niakhar where urogenital schistosomiasis is endemic with seasonal transmission [12,13]. For the first time, we have been able to show the presence of *Schistosoma bovis* in the Niakhar area. We have also shown a probable co-infection of *Schistosoma haematobium*/*Schistosoma bovis* DNA of both species were found within a single snail specimen. These results are similar to those obtained by Pennance et al. in Niger [11]. However, even if RD-PCR had been designed for the diagnosis of *S. haematobium*, *S. mattheei*, and *S. bovis*, the double band profiles for cox1 could also be observed for *S. curassoni* and *S. guineensis* [28]. Therefore, we cannot exclude the possibility that the diagnosis of *S. bovis* was confused with *S. curassoni*, which is known to be present in Senegal [33]. The overlap of two *Schistosoma* species in a single *B. senegalensis* snail would suggest a strong interaction between human and livestock schistosomes, raising questions about the likely presence of hybrids in Niakhar. This adds a level of complexity to the epidemiology of urogenital *Schistosoma* in Niakhar. However, this study does not highlight possible hybridizations, which would first require individualization of the parasites infesting the snails with a genotyping approach [1,33]. Therefore, further molecular studies are needed to characterize the *Bulinus* spp. population in Niakhar; until now, only morphological techniques have been used to determine the species encountered in the area, the complication that *B. umbilicatus* could display slight compatibility with *S. bovis* [34]. Moreover, on the other hand, we may need more powerful and specific molecular tools to better characterize the population of *Schistosoma* as the RD-PCR does not yield a complete identification of the sample because it targets only *Schistosoma haematobium*, *Schistosoma bovis*, and *Schistosoma mattheei*. The remaining negative samples from the RD-PCR could therefore be other species of the *haematobium*-group or hybrids strains. However, although the RD-PCR technique is very sensitive [28], we do not exclude the fact that it may not have amplified some samples positive for Dra 1 which is also known for its high sensitivity [35]. It should also be noted that the small number of samples tested is a limitation and does not further enhance our understanding of the RD-PCR test’s sensitivity.

Thus, sequencing techniques or more specific PCR methods targeting a wide range of schistosome species are needed to gain a better understanding of *Schistosoma* infesting *Bulinus* spp. snails in Niakhar [31]. This identification should be performed on an individual basis by recovering parasite specimens through the cercarial shedding method [36]. Indeed, even if the molecular method were more sensitive and specific, it would not give any indication of the parasites’ infection potential when they are directly recovered from the snail tissue extracts. This suggests that the prevalence data obtained from our molecular analyses should be interpreted carefully because the presence of the parasite’s DNA does not necessarily mean that the snails are infested.

Nevertheless, this study has highlighted that the use of PZQ alone does not effectively control the disease, because our sampling was carried out following three mass-treatment campaigns [13]. Even if the prevalence of disease had decreased within the human population, its persistence in intermediate hosts showed that the threat of transmission had been maintained. Furthermore, several months after we sampled the snails, an epidemic rebound of urinary schistosomiasis occurred; the highest disease prevalence rates were observed in the village of Ngangarlam and Sass, where we recorded the highest snail infestation rates [13]. This shows the need to combine snail control and sanitation programs with mass drug distribution, focusing on seasonal transmission hotspots [37,38]. The study of snail parasite interactions could also help to predict the emergence of transmission foci and epidemiological rebounds following mass PZQ administration (MDA).

In this study, the levels of snail infestation dynamic were not uniform, with higher infestation rates found in the northern area of Niakhar compared to the south. In addition, the levels of snail infestation were very heterogeneous between the villages. Indeed, the highest snail infestation rates were observed in Sass njafaj, Ngangarlam, and Diohin which have one pond each. In contrast in Gadiak and Ngalagne Kop where four ponds are present in each village, lower infestation rates were detected. This shows that the level of transmission does not depend directly or exclusively on the density of ponds and it is likely that the presence of latrines or access to drinking water may explain the lower level of snail infestation in Gadiak and Ngalagne Kop compared to Ngangarlam and Diohin. Further epidemiological studies are needed to determine the risk factors of snail infestation dynamics between the different villages, such as a lack of drinking water and latrines. Similarly, the levels of snail infestation were very heterogeneous at the beginning, middle, and end of the rainy season, and with respect to the village. Therefore, the risk of transmission within the population could be highly variable from one place to another depending on the time of the year. Even though the snail infestations peaked in the middle of the rainy season, in the majority of villages, snails infested with the parasites were found at the beginning of the rainy season in July, as shown by a previous study in this area [12]. This indicates that human populations could be at risk of exposure to schistosome transmission earlier than previously thought. This small-scale heterogeneity of the snail infestation level, both in time and space, could indicate complex dynamics of transmission in humans and should be taken into account to better control schistosomiasis, particularly adopting the mass distribution of PZQ at the village level, whilst considering the morbidity data [39]. However, the heterogeneity highlighted in this study could have been more relevant if all samples collected in the 2013 survey had been tested. In areas of seasonal transmission such as Niakhar, this could help in the development of effective control strategies at the village level.

## 5. Conclusions

For the first time, the use of a molecular tool has highlighted a high *Bulinus* snail infestation rate by *Schistosoma haematobium* in the Niakhar area of Senegal, an endemic focus of urogenital schistosomiasis. The presence of both *S. haematobium* and *S. bovis* suggests the potential of parasite hybridization. Further molecular and genetic studies are needed to identify trematodes present in snails and to decipher the molecular interactions between different parasite species. The spatio-temporal dynamics of snail infestation indicate a heterogeneous risk of disease transmission in the Niakhar area, which must be taken into account in the implementation of control strategies.

## Figures and Tables

**Figure 1 pathogens-10-00860-f001:**
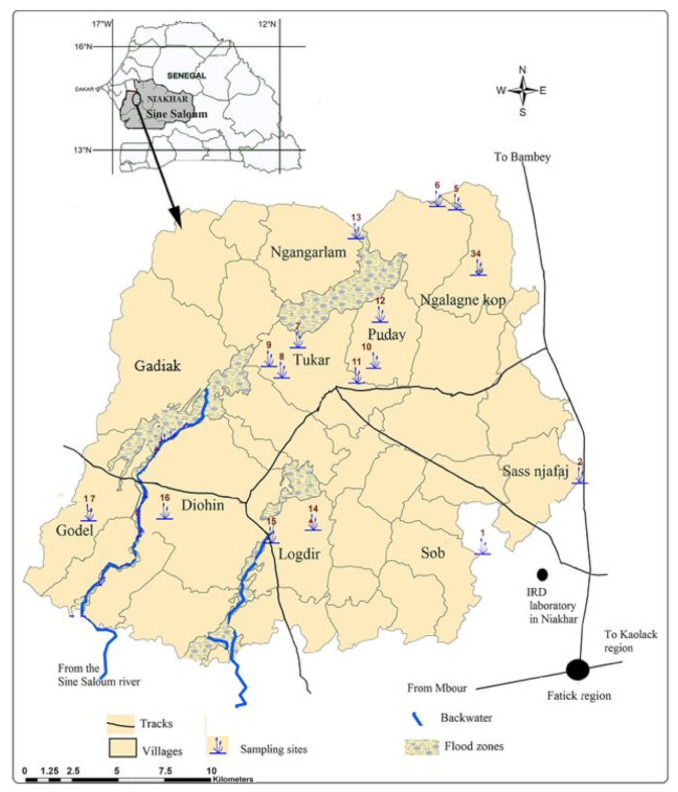
Identification and distribution of different snail collection sites (ponds) in each village of the Niakhar area during the 2013 rainy season [12].

**Figure 2 pathogens-10-00860-f002:**
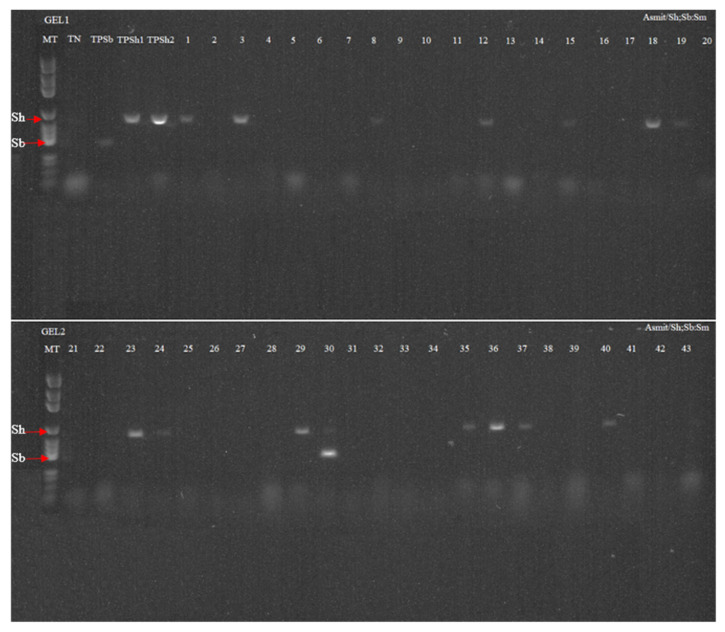
Rapid diagnostic multiplex PCR on *Schistosoma* spp. Amplification reactions of *S. haematobium* genomic DNA are shown at 543 bp (lane 1, lane3, lane 8, lane 12, lane 15, lane 18, lane 19, lane 23, lane 24, lane 29, lane 35, lane 36, lane 37 and lane 40) and the double profile *S. bovis* (Sb)*/S. haematobium* (Sh) (at 306 bp and 543 bp, respectively) (lane 30). NT: negative control, TPSb: positive control *S. bovis*; (TPSh 1 and TPSh 2): Positive control *S. haematobium* and MT: DNA size marker.

**Figure 3 pathogens-10-00860-f003:**
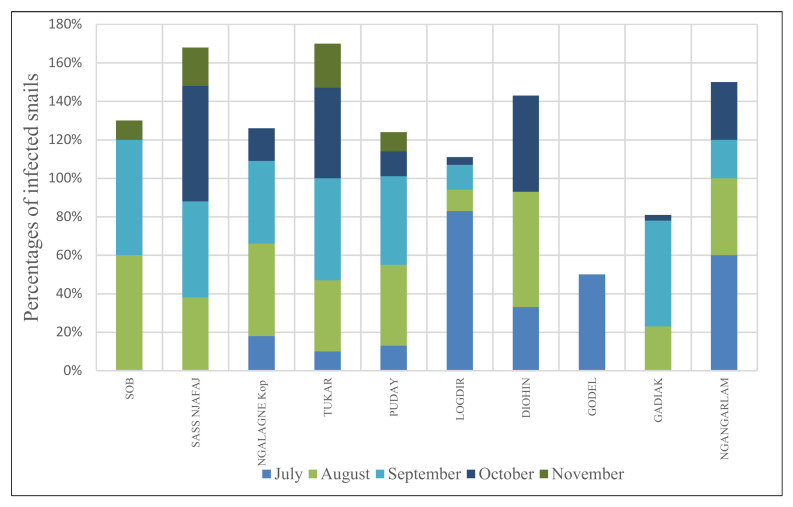
Infestation rate of the *haematobium*-group in snails in each studied village.

**Table 1 pathogens-10-00860-t001:** Total number of snails tested and infestation percentages for each village.

Villages	Number of Snails		
	*B. senegalensis*	*B. umbilicatus*	Infested Snails (%)
Sob	50	-	13 (26.0%)
Sass njafaj	38	-	16 (42.1%)
Ngalagne Kop	156	-	49 (31.4%)
Tukar	130	-	46 (35.4%)
Puday	113	-	28 (24.8%)
Logdir	78	-	12 (15.4%)
Diohin	49	-	14 (28.6%)
Godel	30	-	5 (16.7%)
Gadiak	116	-	32 (27.6%)
Ngangarlam	30	20	21 (42.0%)
Total	790	20	236 (29.1%)

**Table 2 pathogens-10-00860-t002:** Infestation rate by the period of season and villages.

Number of Snails Checked (% Infestation)				
Villages	Beginning of Season (%)	Middle of Season (%)	End of Season (%)	Total (%)
Sob	10 (0.00)	20 (60.00)	20 (5.00)	50 (26.00)
Sass njafaj	0 (0.00)	18 (44.44)	20 (40.00)	38 (42.11)
Ngalagne Kop	39 (17.95)	80 (45.00)	37 (16.22)	156 (31.41)
Tukar	10 (10.00)	60 (40.00)	60 (35.00)	130 (35.39)
Puday	30 (13.33)	43 (44.19)	40 (12.50)	113 (24.78)
Logdir	6 (83.33)	48 (12.50)	24 (4.17)	78 (15.38)
Diohin	9 (33.33)	20 (30.00)	20 (25.00)	49 (28.57)
Godel	10 (50.00)	10 (0.00)	10 (0.00)	30 (16.67)
Gadiak	0 (0.00)	80 (38.75)	36 (2.78)	116 (27.59)
Ngangarlam	20 (60.00)	20 (30.00)	10 (30.00)	50 (42.00)
**TOTAL**	134 (27.61)	399 (37.09)	277 (18.41)	**810 (29.14)**
	*p* = 0.04516	*p* = 2.533 × 10^−7^	---	
χ-squared = 27.816, df = 2, *p* = 9.118 × 10^−7^				

## Data Availability

All relevant data are provided within the manuscripts. Raw data can be made available upon reasonable request.

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
