# Peer review of "Bulinus senegalensis and Bulinus umbilicatus Snail Infestations by the Schistosoma haematobium Group in Niakhar, Senegal"

_pathogens, 2021, doi:10.3390/pathogens10070860_

Round 1

Reviewer 1 Report

This manuscript is looking at species of Bulinus snails in Niakhar, Senegal, and levels of Schistosoma haematobium group species infections within the snails.

Techniques used are qPCR and RD-PCR using Dra1 target. This study is of interest for transmission studies within this area. And important for mapping of snails and schistsomes within Senegal.  

General comments –

  • The language and spelling needs checking throughout.
  • Species names need to be italicised throughout (eg L56 Schistosoma haematobium).
  • Be consistent with spelling and capitalize.

Title: too long – could be “infection status of Bulinus senegalensis and Bulinus umbilicatus in Niakhar…”

Abstract: L19 Reword – this does not make sense.

L22/L25 Dra 1 and DRA1 – be consistent.

L29 are these significant

Introduction

L38 delete “is”

L41 change “Sub-Sahara” to “sub-Sahara”

L60-64 Reword does not make sense

L66 in snails/animals/humans?

Materials and methods

L 81 capitalise months

L82 rain fall per year? Month?

L83 move to molecular analysis

L84 Please give more information on how were the snails collected? Scoop/dredge – how many people? For how long?

How were snails identified? Using shell morphology or molecular methods?

L91 Move to molecular analysis – does not fit here.

L100 titles need to be consistent. And italics for species names.

L101 Summarise or give more information here please. How did you get the head/foot tissue? Why did you only use this part of the snail when the infection could be elsewhere within the snail – eg hepatopancreas

L110 – culot – I’m not sure what this means. Perhaps the DNA was resuspended in 200ul of pure water.

L113 Real-time or quantitative PCR – be consistent and add to acronyms list.

General – make sure primers are cited if you did not design them yourselves.

Results

Capitalise months please.

L188 – 195 very difficult to follow – these data are in table and figure too. Please make all of these more clear.

Discussion

Why was cercarial shedding not performed alongside snail collection? I don’t understand using a previous study. Some of this needs rewriting – to make clearer and some bits moved to the results section.
Also, throughout please check when using bilharzia and schistosomiasis - I would suggest using schistosomiasis throughout.

Figure 1 – no scale bar – please add. Caption is not informative.

Figure 2 – not very clear

Reviewer 2 Report

Reviewer comments: Gaye et al 2021 – Temporal dynamics of B. senegalensis and B. umbilicatus

This study conducted by Gaye et al. investigates the presences of Schistosoma species infecting Bulinus spp. snails in the Niakhar region of Senegal. The investigation into the snail intermediate hosts of schistosomiasis in this area is of interest, and I commend the authors for their work in taking on this challenging subject. However, the manuscript is currently not fit for publication and requires major revision throughout.

Significant improvements need to be made in terms of the formatting, English language, presentation of the methods and results and to provide a relevant discussion that is supported by what is presented in the study. More importantly, the article lacks some insight into the published literature and therefore have inferred some of the results incorrectly (e.g. see comment below on RD-PCR) and don not provide information to the reader.

There is some very useful and informative information here, and I hope the authors can take time to work further on this. I recommend for the authors to redraft and get feedback from researchers in this field of research and also to do so whilst comparing to other manuscripts in this subject area to see what content is missing. Editing from a native English co-author would also improve general readability.

A few extra notes – I started doing line by line but there were too many errors to continue in this fashion.

Suggest shortening the title as currently quite long. Should also be ‘Dynamics’ and use term infection rather than infestation. Title also reads as though only looking at infected snails. Also state country – Niakhar Senegal

Abstract

19 – spp should be followed by a period and should not be in italics – ‘spp.’

20 – PZQ used for praziquantel but not yet defined. Should be defined before use

22 - Dra 1 used and then DRA1 later. Use one.

24 – RD-PCR not defined

25 = should give an n = XX value for the perecentages so that reder can interpret how many B. umbilicatus and B. senegalensis that were positive, as numbers collected not listed yet.

Intro

37 – 38 – ‘it ranks as one..’

40 – 200,000

50 – Schistosoma italics and throughout.

54 – S. haematobium group – formatting consistency for Latin names and groups, acronyms etc. needs major improvement.

*Stopped doing line by line corrections from here as too many to keep listing *

2.1 and 2.2 etc. not formatted correctly i.e. bold / italics? Sectioning confusing. Avoid sub sections of subsections in the methods. The complexity of the study doe not warrant this.

156 – No explanation as to why only 810 of the 8551 snails collected were analysed, were these others not Bulinus? If these were all Bulinus, should maybe demonstrate the proportion of Bulinus from each village that were tested and the total numbers from each. How was the study sample selected? Are the other snails different Genus? How were the snails correctly identified, was morphological guides used?

179 – 180. Almost half of the samples were not amplified by the RD-PCR. Authors provide no valid explanation as to why these are negative (primers used can amplify multiple S. haematobium group species not just Sh and Sb and Sm) and therefore could demonstrate that the Dra1 (as expected) is giving lots of false positives.

Also no explanation as to where the 43 selected samples used in the RD-PCR are from? What villages etc?

If you read the paper by Webster (2010) regarding the Schistosoma RD-PCR for Sh and Sb, you will see that double band profiles for the cox1 can also be observed for S. mattheei, S. curassoni and S. guineensis. S. curassoni is known to be present in Senegal, and therefore the double banding could be explained by this rather than mixed infection. Sequencing the nuclear ITS would have vastly improved the molecular detections here. Note that the ShR and SbR primers used were designed in Webster (2010) and there is more information on the targets of the PCR in this paper, not included in the Schols (2019) later reference used here. This also highlights some incorrect citations.

241 – typo S. umbilicatus

276 – making discussion points about latrines and safe water sources yet neither were investigated or commented on in the study.

Reviewer 3 Report

Please find attached the comments to the authors.

Round 2

Reviewer 2 Report

I can see that after the initial rejection of this manuscript, much work has been done to improve the scientific content of this manuscript, which I am happy to see! However, after just the initial pages of the manuscript I can see there is still a need for extensive English editing and formatting, with errors that should not be present at these final stages of the manuscript. I will request for the editors to send out for full review again once these have been addressed, as only then can the manuscript be properly scrutinised. 
